# A Distributional Robustness Perspective on Adversarial Training with the ∞-Wasserstein Distance

## Abstract

While ML tools are becoming increasingly used in industrial applications, adversarial examples remain a critical flaw of neural networks. These imperceptible perturbations of natural inputs are, on average, misclassified by most of the state-of-the-art classifiers. By slightly modifying each data point, the attacker is creating a new distribution of inputs for the classifier. In this work, we consider the adversarial examples distribution as a tiny shift of the original distribution. We thus propose to address the problem of adversarial training (AT) within the framework of distributional robustness optimization (DRO). We show a formal connection between our formulation and optimal transport by relaxing AT into DRO problem with an ∞-Wasserstein constraint. This connection motivates using an entropic regularizer– a standard tool in optimal transport— for our problem. We then prove the existence and uniqueness of an optimal regularized distribution of adversarial examples against a class of classifier (e.g., a given architecture) that we eventually use to robustly train a classifier. Using these theoretical insights, we propose to use Langevin Monte Carlo to sample from this optimal distribution of adversarial examples and train robust classifiers outperforming the standard baseline and providing a speed-up of respectively $\times 200$ for MNIST and $\times 8$ for CIFAR-10.

## 1 Introduction

We call adversarial example an input which is a human-imperceptible $\epsilon$-perturbation[1] compared to a real example that results in an incorrect classification from a classifier [Goodfellow et al., 2014; Sun et al., 2018; Athalye et al., 2018; Santurkar et al., 2019; Nguyen et al., 2015a; Kurakin et al., 2016; Moosavi-Dezfooli et al., 2016]. The particularity of these examples, which justifies their study, is the fact that most of the so-called adversarial examples are misclassified by a large majority of state-of-the-art neural networks. Long seen as bugs, Santurkar et al. [2019] asserts that the existence of adversarial examples is explained by the presence of easy-to-perturb patterns within the data distribution that are not perceptible by humans but useful for the classification task.

In response to this discovery, different methods have been developed to train robust classifiers and to craft adversarial attacks by determining optimal perturbations [Nguyen et al., 2015b; Papernot et al., 2016a; Goodfellow et al., 2014; Papernot et al., 2016b; Tramèr et al., 2017; Madry et al., 2017]. Among these training methods, adversarial training [Goodfellow et al., 2014; Tramer et al., 2017; Madry et al., 2018] has settled as one of the strongest baselines to train robust classifiers. This method is relatively simple: it consists of training the classifier directly on batches of adversarial examples, leading to the following optimization problem [Madry et al., 2018]:

$$\min_{\theta \in \mathbb{R}^d} \mathbb{E}_{(x,y) \sim p_{data}} \big[ \max_{\|\tilde{x}-x\|_{\infty} \leq \epsilon} \ell(f_{\theta}(\tilde{x}, y)) \big] \tag{1}$$

where $\ell$ is usually the cross-entropy loss and $p_{data}$ is the dataset distribution. Our goal is to rigorously connect adversarial training and robustness by providing a distributional robustness

---

[1]Adversarial example can be defined with respect to any notion of distance that captures the fact that an $\epsilon$-perturbation is imperceptible by a human. In this work, for the sake of simplicity, we focus on the $\ell_{\infty}$ norm, which is the most common one.

perspective [Delage and Ye, 2010; Ben-Tal and Nemirovski, 1998; Sinha et al., 2018] for adversarial training. Distributional robustness is a framework to study predictive models that aim at being robust against distribution shift. Given a set of possible distribution $\mathcal{P}$ that a given predictive model $f_\theta$ could encounter the distributionally robust classification task is:

$$\min_{\theta \in \mathbb{R}^d} \max_{p_{adv} \in \mathcal{P}} \mathbb{E}_{\tilde{x} \sim p_{adv}}[\ell(f_\theta(\tilde{x}, y))] \qquad (2)$$

Note that while standard adversarial training aims to use the point-wise optimal adversarial $\tilde{x}_j$ examples (i.e. for each natural input $x_i$), our perspective considers the optimal adversarial distribution. Working at the scale of distributions leads us to draw inspiration from the Kantorovich relaxation in Optimal Transport theory. To do so, we will define the set $\mathcal{P}$ using coupling measures between natural and adversarial distributions with the help of $\infty$-$\infty$-Wasserstein distance.

**Contributions.** Our contributions are three-fold: *First*, we provide a strong connection between adversarial training and adversarial robustness and distributional robustness by using some tools from optimal transport, we call this formulation *adversarial transport*. *Our second contribution* is to show that, in this specific setting, there exists, against *any* given classifier $f_\theta$, an optimal distribution of adversarial examples and to provide a closed-form solution for this distribution. *Our third contribution* is to use these theoretical insights to come up with a practical training method using Langevin Monte-Carlo sampling to jointly find the optimal classifier and the optimal distribution of adversarial examples. By using this new technique for adversarial training we obtain robust classifiers outperforming the standard baseline [Madry et al., 2018] in terms of robustness and clean accuracy. Moreover, this training technique provides a speed-up of respectively $\times 200$ for MNIST and $\times 8$ for CIFAR in terms of training time.

**Related Work.** Among the various works about defenses and attacks in the context of adversarial examples, Adversarial Training [Madry et al., 2018] is the most common baseline. However, in this procedure, adversarial examples are generated independently of the others by searching for the optimal perturbation for each one. Thus, AT seeks point-wise optimality and not global optimality. Here, the goal is to make our classifier robust to any unknown adversarial example distribution. It justifies the consideration of the DRO framework. DRO enables a mathematical formulation for dealing with uncertainty in complex systems [Delage and Ye, 2010; Lam, 2018; Rahimian and Mehrotra, 2019]. In ML, DRO tries to minimize the loss over the worst-case distribution in a neighborhood of the observed training data distribution [Duchi and Namkoong, 2016; 2018; Chouzenoux et al., 2019; Duchi et al., 2016]. Thus, DRO is pertinent in the case of adversarial examples and allows to bring a principled distributional perspective on AT.

Within the DRO framework, one crucial point is to properly define the uncertainty set which contains the underlying data distribution. The importance of this set lies in the fact that it allows to reduce the computation time and the search area of the underlying distribution [Delage and Ye, 2010; Ben-Tal and Nemirovski, 1998; Lam, 2018]. Different methods have been proposed to construct uncertainty sets. Among them, one considers the help of OT operators and defines the set with all the probability measures such as the $p$-Wasserstein distance from the observed distribution is less than a given constant ($p$-Wasserstein ball)[Blanchet and Murthy, 2019].

Some works have started to study $p$-Wasserstein DRO in the case of adversarial examples. Zheng et al. [2019] defines Distributionally Adversarial Attacks (DAA) which is a generalization of PGD attacks on the space of probability measures with $W_2$-DRO and [Staib and Jegelka, 2017] did the $W_p$-DRO generalization of AT. E.Wong et al. [2019] also considers the $W_p$ ball using an explicit formulation of the uncertainty set with coupling matrices. However, while the previous works modified the AT problem—the $W_p$ constraint on the adversarial distribution does not correspond to the distribution of adversarial examples computed in AT—our work aims at overcoming that. The original AT problem stays a the scale of the example and considers OT to deal with the displacement of pixels. This perspective leads to the discovery that the standard AT problem corresponds to an $\infty$-Wasserstein DRO problem with the $\ell_\infty$ underlying geometry.

In the meantime, as in our paper, entropic regularization is used to find closed forms of the coupling matrix. To extend this understanding of adversarial examples in the case of distributions we generalize this uncertainty set with the $\infty$-$\infty$-Wasserstein distance and add entropic regularization. 2-$\infty$-Wasserstein DRO has already started to receive consideration in the case of AT [Gao et al., 2017].

Nevertheless, the use of the $\ell_\infty$ within the $\infty$-Wasserstein distance allows both to consider the DRO problem at the dataset scale but also to ensure that the adversarial condition is respected for each

element of the adversarial distribution. Together with the use of coupling matrices in our formulation, we found a concave and tractable formulation of our problem which results in a different technique to sample from this distribution (namely Langevin Monte Carlo). Finally, we propose to keep them in memory to reuse the past samples for the update of the robust classifier.

**Notations.** In this work, we will consider a measurable finite dimensional input space $\mathcal{X} \subset \mathbb{R}^d$. We will note $\mathcal{B}(\mathcal{X})$ the Borel $\sigma$-algebra of $\mathcal{X}$ and $\mathcal{M}(\mathcal{X})_+^1$ the set of all probability measures on $\mathcal{X}$.

If not precised, a map $T : \mathcal{X} \to \mathcal{X}$ is considered measurable. We, recall that given a separable metric space $\mathcal{X}$ s.t. any probability measure on $\mathcal{X}$ is a Radon measure, for $T : \mathcal{X} \to \mathcal{X}$, the push-forward measure of $\beta = T_\sharp \alpha \in \mathcal{M}(\mathcal{X})_+^1$ of some $\alpha \in \mathcal{M}(\mathcal{X})_+^1$ satisfies: $\forall h \in \mathcal{C}(\mathcal{X}), \int_\mathcal{X} h(\tilde{x}) d\beta(\tilde{x}) = \int_\mathcal{X} h(T(x)) d\alpha(x)$.

## 2 ADVERSARIAL TRAINING, TRANSPORT, AND DISTRIBUTIONAL ROBUSTNESS

In this work, we say that $\tilde{x} \in \mathcal{X}$ is an adversarial example of $f$ at the data point $(x, y)$ if $\|\tilde{x} - x\|_\infty \leq \epsilon$ [Goodfellow et al., 2014]. Given a loss function $\ell$, an optimal adversarial example $\tilde{x}$ aims at maximizing the following optimization problem:

$$\tilde{x} \in \arg \max_{\|\tilde{x} - x\|_\infty \leq \epsilon} \ell(f_\theta(\tilde{x}), y). \tag{3}$$

In practice, a large value for $\ell(f_\theta(\tilde{x}), y)$ will yield a misclassification of the example $y$. By considering the adversarial training problem (1), one can notice that the distribution of input-output pairs seen by the classifier $f_\theta$ depends on the value of its own parameters. We will call this distribution $p_{adv}$. Such a distribution can be described as the pushforward of a transport map $T$ that respect the adversarial example constraint $\|T(x) - x\|_\infty \leq \epsilon$, $\forall x \in \mathcal{X}$. In other words we have,

$$p_{adv} = T_\sharp p_{data} : (\tilde{x}, y) \sim p_{adv} \iff \exists (x, y) \sim p_{data} \text{ s.t. } \tilde{x} = T(x) \text{ and } \|\tilde{x} - x\|_\infty \leq \epsilon. \tag{4}$$

This remark leads us to our first proposition that explicitly proposes a distributional robust optimization (DRO) formulation of the adversarial training problem.

**Proposition 2.1** (DRO formulation of adversarial training). *The adversarial training problem* (1) *can be formulated as a DRO problem* (2) *where:*

$$\mathcal{P} = \{p_{adv} : \exists T : \mathcal{X} \to \mathcal{X}, \ p_{adv} = T_\sharp p_{data} \ and \ \|T(x) - x\|_\infty \leq \epsilon, \forall x \in \mathcal{X}\} \tag{5}$$

Interestingly, one can draw parallels with Monge's formulation of optimal transport (OT), were given an initial measure $\mu$ and a target measure $\nu$, the task is to solve the following problem

$$\inf_T \int_\mathcal{X} c(x, T(x)) d\mu(x) \quad \text{subject to} \quad \nu = T_\sharp \mu. \tag{6}$$

On the one hand, (6) and the DRO problem defined in Proposition 2.1 are drastically different problems in essence. It can be explained by the difference between the constraints on the transportation map and the fact that while Equation 6 aims at minimizing a function of the transportation map $T$, Equation 2 aims at maximizing a function of the transported distribution $p_{adv}$.

On the other hand, it is insightful to notice how both problems depend on a set of transportation maps to connect the initial and the target distributions.

The limitations of Monge's problem are well known in the OT community: due to its non-linear constraint, the problem is often intractable and may not have a solution (i.e. the infimum may not be achieved). That is why Kantorovich [1940], proposed to relax (6) into a problem with linear constraints by considering coupling measures (an approach that allows a mass splitting from a natural example $x_i$ to different locations $\tilde{x}_j$). Such a relaxation had the advantage to provide a better understanding of the original problem leading to novel theoretical guarantees (e.g., the existence of solutions for Kantorovich's problem, duality, and in some situations equality between Monge's problem and Kantorovich's problem) as well as tractable algorithms [Sinkhorn, 1964; Cuturi, 2013b].

In a similar vein, we believe our perspective on adversarial examples that combine ideas from distributional robustness and adversarial transport leads to a better understanding of the problem of adversarial training. Thus, we will use that insight to consider coupling measures to relax the adversarial training problem (1). It will lead to a new DRO formulation for adversarial training that we call *adversarial transport*.

## 3 ADVERSARIAL TRANSPORT FORMULATION

We first define the notion of coupling measure, that we will use to define a relaxed version of the DRO formulation described in Proposition 2.1.

**Definition 3.1** (Transport plan). *Let $p_{adv}$ and $p_{data}$ be two distributions on $\mathcal{X}$. $\pi \in \mathcal{B}(\mathcal{X}) \otimes \mathcal{B}(\mathcal{X}) \to [0,1]$ is a transport plan between $p_{adv}$ and $p_{data}$ if:*

$$\forall A \in \mathcal{B}(\mathcal{X}), \ \pi(A \times \mathcal{X}) = p_{data}(A) \quad and \quad \forall B \in \mathcal{B}(\mathcal{X}), \ \pi(\mathcal{X} \times B) = p_{adv}(B) \tag{7}$$

*The space of transport plan between $p_{data}$ and $p_{adv}$ is denoted $\Pi(p_{data}, p_{adv})$.*

Informally, the coupling measure $\pi : \mathcal{B}(\mathcal{X}) \otimes \mathcal{B}(\mathcal{X})$ describes the spatial modification of the initial distribution $p_{data}$ to the adversarial distribution $p_{adv}$. For instance, in the discrete case where we have the datapoints $(x_i)$ and the potential adversarial examples $(\tilde{x}_j)$, a coupling is a matrix $P$ such that the value $P_{i,j}$ represents the amount of mass flowing from the natural example $i$ to the adversarial example $j$.

### 3.1 INCORPORATING THE PROXIMITY CONSTRAINT TO THE COUPLING.

Adversarial examples are tied to the original examples by a proximity constraint. Thus, a DRO formulation of adversarial training must encompass such a constraint. For a coupling measure it corresponds to the fact that it can transport $(x, y) \sim p_{data}$ to $(\tilde{x}, y) \sim p_{adv}$ only if $\|x - \tilde{x}\|_\infty \leq \epsilon$.

**Proposition 3.1** (DRO relaxation of adversarial training). *The DRO problem* (2) *can be relaxed by considering coupling measure. In that case the constraint set $\mathcal{P}$ in Equation 2 take the form:*

$$\mathcal{P}_{conv} = \{p_{adv} \ : \ \exists \pi \in \Pi(p_{data}|p_{adv}) \ \text{ with } \ \text{supp}(\pi) \subset \{(x, y, \tilde{x}, y) \ | \ \|\tilde{x} - x\|_\infty \leq \epsilon, \ y \in \mathcal{Y} \ \}\}.$$

This proposition proposes a convex relaxation of the initial constrain set (5). Intuitively, the set $\mathcal{P}$ constraints the distribution $p_{adv}$ to be $\epsilon$-close to $p_{data}$. Because this set is convex we can hope it corresponds to a notion $\epsilon$-ball. We show in the next subsection that this intuition can be formalized by using the $\infty$-Wasserstein distance.

### 3.2 CONNECTION WITH $\infty$-WASSERSTEIN

Given two distributions $\mu$ and $\nu$ and a distance function $d$, the $p$-$p'$-Wasserstein distance corresponds to the minimal average $\ell_p$ norm to the power of $p$ over the transport plans between $\mu$ and $\nu$. Formally, by calling $z = (x, y)$ and $\mathcal{Z} := \mathcal{X} \times \mathcal{Y}$, it can be defined as

$$W_{p,p'}(\mu, \nu) := \inf_{\pi \in \Pi(\mu,\nu)} \Big( \int_{\mathcal{Z} \times \mathcal{Z}} \|z - \tilde{z}\|_p^{p'} d\pi(z, \tilde{z}) \Big)^{1/p'}. \tag{8}$$

The $p$-$\infty$-Wasserstein distance can be defined as the limiting case $p' \to \infty$. Thus, it can be seen as the minimal amount of point-wise transport (w.r.t. the $\ell_p$ norm) necessary to transport $\mu$ to $\nu$,

$$W_{p,\infty}(\mu, \nu) = \inf_{\pi \in \Pi(\mu,\nu)} \sup_{(z,\tilde{z}) \in \text{supp}(\pi)} \|z - \tilde{z}\|_p. \tag{9}$$

Thus by considering $p = \infty$, we can establish that the set introduced in Proposition 3.1 can be described using the $\infty$-$\infty$-Wasserstein distance.

**Proposition 3.2.** *[convex relaxation of adversarial training]For any $\epsilon < 1$,[2] the convex DRO relaxation of adversarial training can be reformulated as*

$$\min_{\theta \in \mathbb{R}^d} \max_{p_{adv} \in \mathcal{B}_\epsilon(p_{data})} \mathbb{E}_{(\tilde{x},y) \sim p_{adv}}[\ell(f_\theta(\tilde{x}, y)] \tag{10}$$

*where $\mathcal{B}_\epsilon(p_{data}) = \{p \mid W_{\infty,\infty}(p, p_{data}) \leq \epsilon\}$ and $W_{p,\infty}$ is defined in* (9).

In other words, adversarial training can be seen as a DRO problem restricted to the distributions that are $\epsilon$-close to the data distribution with respect to the $\infty$-$\infty$-Wasserstein ($\infty$-Wasserstein distance within a space with the $\ell_\infty$ metric).

The $p$-$p'$-Wasserstein distance has been widely studied in the context of optimal transport [Champion et al., 2008; Villani, 2003]. The closeness of our formulation with OT encourages us to add an entropic regularization [Peyré and Cuturi, 2019; Cuturi, 2013a].

---

[2]This assumption is only for simplicity of exposition, what we need in general is that $\epsilon < \min_{y \neq y'} |y - y'|$ which can always be achieved by reparametrizing the discrete labels.

### 3.3 ENTROPY-REGULARIZED FORMULATION

Entropic regularization has been a very successful approach for finding an approximate solution to OT problems [Cuturi, 2013b; Genevay et al., 2016]. From a theoretical point of view, the entropic regularization prevents the sparsity of the coupling matrix, which is justified from a practical aspect in the context of transport by the fact that in practice the transport network is diffuse [Wilson, 1969]. In the context of adversarial examples, this is justified by the fact that the adversarial examples are mostly located in areas of low probability for the input distribution [Song et al., 2017; Ma et al., 2018]. We suppose that $p_{data}$ admits a density. For an arbitrary coupling measure $\pi$, we can define a relative entropy function as $H(\pi) = -\int_{\mathcal{Z}\times\mathcal{Z}}(\log(\pi(z,\tilde{z})) - 1)d\pi(z,\tilde{z})$ with $H(\pi) = +\infty$ is $\pi$ does not admit a density. Thus our entropy-regularized problem is:

$$\min_{\theta\in\mathbb{R}^d} \max_{\substack{\pi\in\Pi(p_{data},p_{adv})\\ s.t.\ p_{adv}\in\mathcal{B}_\epsilon(p_{data})}} \mathbb{E}_{(\tilde{x},y)\sim p_{adv}}[\ell(f_\theta(\tilde{x}),y)] + \frac{1}{\lambda}H(\pi)\,. \tag{11}$$

The objective is strictly concave in $\pi$, therefore our regularized problem becomes strongly concave which ensures the existence and uniqueness of an optimal solution. It will allow us to obtain an explicit formulation for the optimal coupling $\pi$ and then for the optimal adversarial distribution $p_{adv}$. The solution of this new problem will be an approximate solution of the original problem (A). By calling the optimal coupling $\pi^*_\lambda$ associated to this regularized problem (11) we have the following property:

**Proposition 3.3.** *If $\lambda \to \infty$, then $\pi^*_\lambda \to \pi^*$.*

### 3.4 SOLUTIONS TO THE ADVERSARIAL TRANSPORTATION PROBLEM

In this section, we study the general framework where the two measures $p_{data}$ and $p_{adv}$ are defined on metric spaces $\mathcal{X}$. No particular assumption is made on the form of $p_{data}$ and $p_{adv}$. The inner-objective of penalized adversarial transportation problem is the following:

$$g(\theta) = \max_\pi h_\theta(\pi) := \max_{\substack{\pi\in\Pi(p_{data},p_{adv})\\ s.t.\ \exists p_{adv}\in\mathcal{B}_\epsilon(p_{data})}} \mathbb{E}_{(\tilde{x},y)\sim p_{adv}}[\ell(f_\theta(\tilde{x}),y)] + \frac{1}{\lambda}H(\pi)\,, \tag{12}$$

where $\mathcal{B}_\epsilon(p_{data}) = \{p \mid W_{\infty,\infty}(p,p_{data}) \leq \epsilon\}$. For any given $\theta \in \mathbb{R}^d$, the function $h_\theta$ is strictly-concave and the constraints on $\pi$ are convex (cf Appendix A for proof). Then, by using convex duality, we can deduce that the problem (12) has a unique closed-form solution:

**Proposition 3.4.** *For any given $\theta \in \mathbb{R}^d$, The optimization problem (12) has a unique solution,*

$$\pi^*((x,y),(\tilde{x},\tilde{y})) \propto p_{data}(x,y)\exp(\lambda\ell(f_\theta(\tilde{x}),y))\mathbf{1}_{\|x-\tilde{x}\|_\infty\leq\epsilon}\mathbf{1}_{y=\tilde{y}}\,. \tag{13}$$

*For a given natural example $x$ with a label $y$, $\pi((x,y),(\tilde{x},\tilde{y}))$ grows with the propensity of $\tilde{x}$ to be better than the other elements of $\mathcal{B}_x(\epsilon)$ (the $\ell_\infty$-ball of radius $\epsilon$) to fool the classifier. We then deduce the explicit form of the adversarial distribution:*

$$p^*_{adv}(\tilde{x},y) = \mathbb{E}_{x\sim p_{x,data}}[p_{data}(y|x)\frac{\exp(\lambda\ell(f_\theta(\tilde{x}),y))\mathbf{1}_{\|x-\tilde{x}\|_\infty\leq\epsilon}}{\int_{\tilde{x}\in\mathcal{B}_\epsilon(x)}\exp(\lambda\ell(f_\theta(\tilde{x}),y))d\tilde{x}}] \tag{14}$$

From this proposition, we notice that if $\tilde{x} \in \mathcal{B}_x(\epsilon)$, then $\pi(z,\tilde{z}) > 0$, i.e. any examples which satisfies the adversary condition has a non-zero probability to be considered as an adversarial example of $x$. However, when $\lambda$ is large enough, most of the mass of the distribution lies where $\ell(f_\theta(\tilde{x}),y)$ is close to its maximum value. Moreover it is interesting to notice that the mass of $p_{adv}(\tilde{x},y)$ can come from the transport of *several input data-points* as a potentially infinite number of input data-point $x$ verify the constraint $\mathbf{1}_{\|x-\tilde{x}\|_\infty}$ and thus could be transported to $\tilde{x}$. We will seen in the next section that is never happens in practice.

As we noticed before, the inner maximization problem is strictly-concave in $\pi$ and the constraints are convex. By strong duality, we know that the dual form is equal to the primal. The primal can be expressed with a $\min$ over a lagrange multiplier $\alpha$ s.t. $\alpha(x,y) = -\frac{1}{\lambda}\log(\frac{p_{data}(x,y)}{\int_{\tilde{x}\in\mathcal{B}_\epsilon(x)}\exp(\lambda(\ell(f_\theta(\tilde{x}),y)))d\tilde{x}})$ (c.f. Appendix A for proof) and is convex w.r.t. this variable. Thus, we can express the original min-max optimization problem as a convex minimizaion problem in $\theta$ which leads to our central theorem.

**Theorem 3.1** (Characterization of the solutions). *For any $\lambda > 0$, the solution set of the regularized adversarial training problem* (11) *is:*

$$\theta^* \in \arg\min \mathbb{E}_{(x,y)\sim p_{data}}[\frac{1}{\lambda}\log(\int_{||\tilde{x}-x||\leq\epsilon} \exp(\lambda\ell(f_\theta(\tilde{x}),y))d\tilde{x})] =: \arg\min g(\theta)\,, \qquad (15)$$

*and the optimal adversarial distribution associated with this adversarial training problem is*

$$p_{adv}(\tilde{x},y) = \mathbb{E}_{x\sim p_{x,data}}[p_{data}(y|x)\frac{\exp(\lambda\ell(f_{\theta^*}(\tilde{x}),y))\mathbf{1}_{\|x-\tilde{x}\|_\infty\leq\epsilon}}{\int_{\tilde{x}\in\mathcal{B}_\epsilon(x)}\exp(\lambda\ell(f_\theta(\tilde{x}),y))d\tilde{x}}] \qquad (16)$$

In order to solve (12) we would need to sequentially sample from $\pi$ to compute the gradient of $g$ (see §4) for the exact derivation of $\nabla g(\theta)$. Because of its normalization constant, the computation transport plan (29) for any given value of $\theta$ may be expensive to compute. However we will see in the next sections that: a) in practice, one computes this plan simultaneously with the training on the classifier and b) it has a simpler form in practice when dealing with a finite dataset.

**A Zero-Sum game perspective on Adversarial Transport.** One perspective on (11) is that it corresponds to game between a classifier $f_\theta$ and a transport plan $\pi$ that aims at transporting the data distribution $p_{data}$ into an adversarial distribution $p_{adv}$ that is $\epsilon$-close to $p_{data}$ with respect to the $\infty$-$\infty$-Wasserstein distance. While the classifier aims at minimizing the average cross-entropy loss over the adversarial dataset the transport plan aims at maximizing the errors of the classifier and its own entropy. This perspective is closely related to Adversarial examples games (AEG) [Bose et al., 2020] where the authors tried to learn an adversarial distribution with a generator conditioned on the input examples. This work differs from AEG by its perspective (we connect adversarial examples with DRO), the problem formulation (we look for a regularized transport plan while AEG focused on a generator), the algorithmic implementation (we consider Langevin to find the transport plan), and the finale focus since AEG concentrated on the generation of transferable adversarial example while our focus is on training robust classifiers.

This perspective also gives us an insight: the optimal distribution of adversarial examples may provide adversarial attacks that transfer well across models. The latter point can be supported by the following minimax theorem that is a simple extension of the result from [Bose et al., 2020, Prop. 1].

**Proposition 3.5.** *Let us call $\mathbb{E}_{(\tilde{x},y)\sim p_{adv}}[\ell(f_\theta(\tilde{x}),y)] + \frac{1}{\lambda}H(\pi) = \varphi(\theta,\pi)$. If $\{f_\theta \mid \theta \in \Theta\}$ is a convex set, then we have that*

$$\min_{\theta\in\mathbb{R}^d} \max_{\pi\in\mathcal{P}} \varphi(\theta,\pi) = \max_{\pi\in\mathcal{P}} \min_{\theta\in\mathbb{R}^d} \varphi(\theta,\pi) \qquad (17)$$

$\mathcal{P}_{conv} = \{p_{adv} : \exists\pi \in \Pi(p_{data}|p_{adv}) \text{ with } \text{supp}(\pi) \subset \{(x,y,\tilde{x},y) \mid \|\tilde{x}-x\|_\infty \leq \epsilon, y\in\mathcal{Y}\}\}\,.$

The assumption that the class of function $\{f_\theta \mid \theta \in \Theta\}$ is convex is not always true in practice but it holds in the case of linear classifier or when one assumes that we have infinite capacity classifiers (i.e. we can achieve a measurable function). Moreover, considering the class of neural networks, it has been argued that they were spanning a set that is almost convex [Gidel et al., 2021].

## 3.5 PRACTICAL CASE: SEMI-DISCRETE ADVERSARIAL TRANSPORTATION PROBLEM

In practice, we do not have access to the true distribution $p_{data}$ but only to a finite dataset sampled from this distribution (e.g MNIST/ CIFAR-10, etc). This can be seen as considering the empirical distribution $p_{data} = \frac{1}{n}\sum_i^n \delta_{x_i}$. However, even when considering empirical distributions, the distribution of adversarial examples $p_{adv}$ is an arbitrary measure on metric space $\mathcal{X}$ as an adversarial example could be any point in the $\ell_\infty$-ball of radius $\epsilon$ ball from the initial point (for datapoint $x_i$ we call that ball $\mathcal{B}_\epsilon(x_i)$). Its analysis is thus similar to the continuous case. However, in such a practical case the optimal transport map has a simpler form under the following assumption

**Assumption 3.1** (Empty intersection). *We have that $\mathcal{B}_\epsilon(x_i) \cap \mathcal{B}_\epsilon(x_j)$ for all $1 \leq i \neq j \leq n$.*

For real world dataset, this assumption is almost always true. It means that two datapoints are not $\epsilon$ close which would mean that they are perceptually indistinguishable. We can actually show that if the inputs are high dimensional Assumption 3.1 hold with an exponentially high probability,

**Proposition 3.6.** *Let us assume that the datapoints $x_i$ are independently sampled from a distribution $p$ over $[0,1]^d$ that is absolutely continuous with respect to Lesbegue measure and has a upper-bounded value, i.e. $p(x) \leq M$, $x \in [0,1]^d$. Then $\mathbb{P}(\exists i \neq j \in [n], \|\tilde{x}_j - x_i\|_\infty \leq \epsilon) = O(n^2\epsilon^d)$.*

such a proposition could be extended to the more challenging setting where the data distribution is supported by a manifold but this is out of the scope of this paper since its purpose is purely illustrative.

**Theorem 3.2.** *[Characterization of the solutions in the semi discrete case] For any $\lambda > 0$, the solution set of the regularized adversarial training problem* (11) *is:*

$$\theta^* \in \arg\min \frac{1}{n\lambda} \sum_{i=1}^{n} \log \left( \int_{||\tilde{x}-x_i|| \leq \epsilon} \exp(\lambda \ell(f_\theta(\tilde{x}), y_i)) d\tilde{x} \right) =: \arg\min g(\theta), \quad (18)$$

*and under Assumption 3.1 the optimal adversarial distribution associated with this adversarial training problem is*

$$p_{adv}(\tilde{x}, y) = \begin{cases} \dfrac{\exp(\lambda \ell(f_\theta(\tilde{x}), y))}{n \int_{\tilde{x} \in \mathcal{B}_\epsilon(x_i)} \exp(\lambda \ell(f_\theta(\tilde{x}), y)) d\tilde{x}} & \text{if } \exists i \in [n] \text{ s.t. } \|x_i - \tilde{x}\|_\infty \leq \epsilon \\ 0 & \text{otherwise.} \end{cases} \quad (19)$$

The practical insight from this result is that the distribution of adversarial example can, in practice be generated example-wise. Meaning that for each datapoint $x_i$, $i \in [n]$ one can generate the distribution of adversarial examples around $x_i$ distributed as $p_{adv}(\tilde{x}, y) \propto \exp(\lambda \ell(f_\theta(\tilde{x}), y)), \forall \tilde{x} \|\tilde{x} - x_i\| \leq \epsilon$. Practically, we will maintain a finite dataset of the adversarial examples $(\tilde{x}_{i,k})$ associated with the datapoint $x_i$.

## 4 ADVERSARIAL TRANSPORT ALGORITHM

We want to solve the minimization problem (40). We can solve this problem using gradient descent:

$$\theta^{t+1} = \theta^t - \eta \nabla_\theta g \quad \text{where} \quad \nabla_\theta g = \frac{1}{n} \sum_{i=1}^{n} \int_{B_\epsilon(x_i)} \nabla_\theta \ell(f_\theta(\tilde{x}), y_i) \frac{e^{\lambda \ell(f_\theta(\tilde{x}), y_i)}}{\int_{B_\epsilon(x_i)} e^{\lambda \ell(f_\theta(\tilde{x}), y_i)} d\tilde{x}} d\tilde{x} \quad (20)$$

To evaluate this gradient we need to sample from the distribution $\tilde{q}(\tilde{x}) \propto \exp(\lambda \ell(f_\theta(\tilde{x}), y_i))$ defined over $B_\epsilon(x_i)$. This can be done via Langevin Monte Carlo [Durmus and Moulines, 2016] as the next section explains.

### 4.1 LANGEVIN MONTE CARLO

Unadjestued Langevin Algorithm [Durmus and Moulines, 2016] is convenient to sample high-dimensional probability distribution $q$. Having the following density w.r.t. the Lesbegue measure $q(\tilde{x}) \propto \exp(-U(\tilde{x}))$, one can show that $q$ is the stationnary measure of the following Stochastic Differential Equation:

$$X_{t+1} = X_t - \gamma_{t+1} \nabla U(X_t) + \sqrt{2\gamma_{t+1}} \xi_{t+1} \quad (21)$$

With $(\xi_t)_t$ i.i.d $\mathcal{N}(0, I_d)$ and $(\gamma_t)_t$ constant or decreasing towards 0. $L$-smoothness of $U$ gives existence and uniqueness of a solution. Langevin MC can be seen as the sampling analog of gradient descent in optimization. In this work, we consider a projected version of the Langevin algorithm since we need to sample within some $\ell_\infty$ balls. Such an algorithm has been analyzed by Bubeck et al. [2018] in the case where $U(x)$ is convex. In our practical setting we will consider $U_i(\tilde{x}) = -\lambda \ell(f_\theta(\tilde{x}), y_i))$ for $\tilde{x} \in B_\epsilon(x_i)$, $i \in [n]$ which are non-convex function. However, some results have been proven in that context by Ma et al. [2019]. Moreover, the Langevin algorithm has been recently used with great success in the context of diffusion models for deep learning for which the function $U$ is not convex (see [Weng, 2021] for an overview).

### 4.2 PRACTICAL SETTING

As described in Algorithm 1, the training process has two loops:

1. Given a minibatch $(x_i, y_i)_{i \in B}$, the inner loop samples the adversarial distributions $q_i(x)$ against the current classifier $f_\theta$ via Langevin MC where each samples are then projected onto $\mathcal{B}_i(\epsilon)$ (operator $P_{\mathcal{B}_i(\epsilon)}$) and onto the space $\mathcal{X}$ (operator $P_\mathcal{X}$). It allows us to estimate $\nabla g$.

2. The outer loop consists of the stochastic gradient descent step (20).

**Sign of the Gradient.** Similarly to standard adversarial training and attack methods we use *the sign of the gradient* of $g$ to update $\theta$. It can be motivated by the fact that the sign of gradient is well suited to the geometry defined by the $\ell_\infty$ ball since it corresponds to the steepest descent direction [Boyd et al., 2004] in the $\ell_\infty$ geometry [Balles et al., 2020].

**Reusing the Examples.** In practice, we restart the Langevin MC *from the adversarial examples found at the previous iterations*, this allow us to only use one step of Langevin MC in the inner loop and significantly speed-up training. This idea is justified by the fact that the distribution $p_{adv}(\tilde{x}, y) \propto \exp(\lambda \ell(f_\theta(\tilde{x}), y))$ is actually a continuous function of $\theta$. It can be seen with duality argument from (12) where the function $h_\theta(\pi)$ is *strictly concave* with respect to $\pi$ which implies the continuity of $\theta \mapsto \arg\max_\pi h_\theta(\pi)$. We leave as a future work the formalization of this argument.

## 5 EXPERIMENTAL RESULTS

We investigate the Adversarial Transport Algorithm at training robust classifiers (for a given hypothesis class) against adversarial attacks on MNIST and CIFAR-10.

**Experimental setup** The experiments are realized on MNIST and CIFAR-10. We perform all attacks and training with $\epsilon = 0.3$ for MNIST and $\epsilon = 8.0$ for CIFAR-10 (w.r.t. to the $\ell_\infty$ norm). We train our robustly trained classifier using stochastic gradient descent with the cross-entropy loss. We use a learning rate $l_r = 0.1$ in the case of Adversarial Transport and update $\theta$ with the sign of the gradient.

---

**Algorithm 1** Adversarial Transport with Langevin

1: **input:** dataset $(x_i, y_i)_{i=1}^n$, step-size: $\eta$, number of adversarial examples: $K$, noise variance $\sigma$.
2: Initialization: $\tilde{x}_{i,k} = x_i$, $i \in [n]$, $k \in [K]$
3: **for** $n$ : nb of steps **do**
4:    Sample a minibatch: $(x_i, y_i)_{i \in B}$, $B \subset [n]$
5:    Load the attacks: $\tilde{x}_{i,k}^{(0)} \leftarrow \tilde{x}_{i,k}$, $i \in B, k \in [K]$
6:    **for** $T$ : nb langevin iter **do**
7:      **for** $i \in B, k \in [K]$ **do**
8:       $\tilde{x}_{i,k}^{(t+1)} \leftarrow \tilde{x}_{i,k}^{(t)} + \eta \, \mathrm{sign}(\nabla_x \ell(f_\theta(\tilde{x}_{i,k}^{(t+1)}, y_i))$
9:       $\tilde{x}_{i,k}^{(t+1)} \leftarrow P_{\mathcal{B}_i(\epsilon)}[\tilde{x}_{i,k}^{(t+1)} + \sigma\xi]$
10:      $\tilde{x}_{i,k}^{(t+1)} \leftarrow P_\mathcal{X}[\tilde{x}_{i,k}^{(t+1)}]$
11:    $\nabla_\theta g = \frac{1}{|B|K} \sum_{i \in B, k \in [K]} \nabla_\theta \ell(f_\theta(\tilde{x}_{i,k}^{(t+1)}, y_i))$
12:    $\theta \leftarrow \theta - \eta \nabla_\theta g$
13:    Save the attacks: $\tilde{x}_{i,k} \leftarrow \tilde{x}_{i,k}^{(T)}$, $i \in B, k \in [K]$
14: **Return:** $\theta$

---

For both datasets, we train our classifier with 100 epochs, a batch size of 100. For Langevin MC parameters, we take $K = 1, \eta_l(\lambda) = 0.2$ and $\sigma = 0.2$ (i.e. $\lambda = \frac{2\eta_l}{\sigma^2} = 10$). In theory, we must take $\lambda$ big enough to have a good approximate of the solution. Appendix B.1 shows that in practice $\lambda = 10$ is the best value to ensure convergence of our algorithm.

**Baselines** We test our robust classifier against first-order attacks: Fast Gradient Sign Method (FGSM) attacks [Goodfellow et al., 2014; Szegedy et al., 2014], Projected Gradient Descent (PGD) attacks with 40 iterations (PGD 40) and PGD attacks with 100 iterations (PGD 100) [Madry et al., 2017; 2018]. We compare those results with the performance of classifiers adversarially trained with PGD [Goodfellow et al., 2014; Madry et al., 2017; 2018] on the same attacks (we trained PGD classifiers according to the method described in the paper and use the same parameters: $l_r = 0.01$, 100 epochs, trained with PGD 40). All the attacks are generated via the library Advertorch of Pytorch[3]. The attacks are $\ell_\infty$ bounded. Additional details concerning the tuning of hyperparameters and hypothesis classes used are given in the Appendix B.1[4]. We evalute white-box attacks (i.e. we have access to the gradient of the model we want to attack). The results are provided below:

### 5.1 PERFORMANCE ON MNIST

We first compare both methods on MNIST, the architecture is the same for both frameworks and is described in the Appendix B.2. We make two main observations, first training the model using Langevin improves the overall robustness of the classifier against all attackers we tested against as shown in Table 1. Secondly, by re-using the samples from the previous iterations we converge much

---

[3]https://github.com/BorealisAI/advertorch.git
[4]Code : https://anonymous.4open.science/r/Adversarial-Training-BCA4

faster, indeed for each update of the classifiers we only need to do one back-propagation instead. This leads to a very important speed up as observed in Fig 2.

| Performance | Langevin | PGD |
|---|---|---|
| Train Adversarial | 98.48% | 94.81% |
| Clean test set | 99.16% | 98.82% |
| FGSM | 96.53% | 94.95% |
| PGD 40 | 93.12% | 92.28% |
| PGD 100 | 91.87% | 91.18% |
| Autoattack | 88.55% | 89.10% |

Table 1: Comparison of the performance for MNIST of robust classifiers trained with Langevin and PGD (with 40 iterations). The architecture of the two classifiers is architecture A. The attacks are FGSM, PGD 40, PGD 100, and autoattack (Croce and Hein [2020]). The classifier trained with Langevin is more robust than the one trained with PGD for each measure except Autoattack.

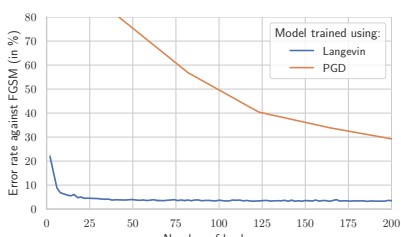

Figure 1: Comparison of Langevin and PGD adversarial training on MNIST. We plot the error rate against FGSM as a function of the number of backprops. We observe that Langevin converges much faster as it only needs to compute 1 gradient per update of the classifier by being able to re-use the samples from the previous iterations. After 8 iterations, Langevin achieves a score that is not even reached by PGD after 1600 iterations.

## 5.2 PERFORMANCE ON CIFAR-10

| Performance | Langevin | PGD |
|---|---|---|
| Train Adversarial | 99.82% | 99.38% |
| Clean test set | 78.72% | 68.93% |
| FGSM | 53.40% | 45.58% |
| PGD 40 | 49.57% | 44.70% |
| PGD 100 | 49.50% | 43.74% |

Table 2: Comparison of the performance for CIFAR-10 of robust classifiers trained with Langevin and PGD (with 40 iterations). The architecture of the two classifiers is RESTNET18. The attacks are FGSM, PGD 40, and PGD 100. As for MNIST, the classifier trained with Langevin is much more robust than the one trained with PGD for each measure.

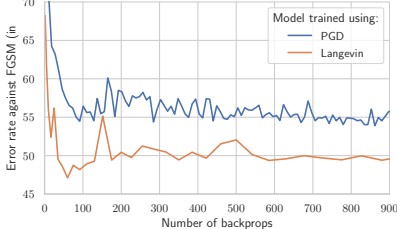

Figure 2: Comparison of Langevin and PGD adversarial training on CIFAR. We plot the error rate against FGSM as a function of the number of backprops. We observe that Langevin converges much faster as it only needs to compute 1 gradient per update of the classifier by being able to re-use the samples from the previous iterations.

## 6 CONCLUSION

In this paper, we introduce the Adversarial Transport Framework. We derived this framework from Adversarial Training relaxation through regularized Distributionally Robust Optimization (DRO) with $\infty$-$\infty$-Wasserstein distance. The outcome of this work provides a close form of the optimal adversarial distribution against a given classifier which can be sampled via Langevin Monte-Carlo. From that, we have designed an algorithm that optimally trains (for a given hypothesis class) a classifier against adversarial examples. The major asset of our work, which surely justifies these outstanding results, is that it considers the problem at the scale of the entire dataset and not at the scale of the example.

Our algorithm brings several novelties: at each iteration, we perform a gradient ascent on the distribution and a gradient descent on the classifier. For this, we keep in memory the current distribution of adversarial examples and iterate on it at each step. In addition, only one iteration of Langevin MC is necessary at each step. As a result, our experiments show that the Adversarial Transport classifier outperforms PGD trained classifier on MNIST and CIFAR-10, but also that our algorithm allows us to train it much faster than with PGD thanks to the novelties described above.

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

## A    PROOFS OF THEORETICAL RESULTS

**Proposition 2.1.** *The adversarial training problem* (1) *can be formulated as a DRO problem* (2) *where:*

$$\mathcal{P} = \{p_{adv} \; : \; \exists T : \mathcal{X} \to \mathcal{X} \, , \; p_{adv} = T_{\sharp} p_{data} \;\; and \;\; \|T(x) - x\|_{\infty} \leq \epsilon \, , \; \forall x \in \mathcal{X}\} \tag{22}$$

*Proof of Proposition 2.1.* Let us start with the original adversarial training problem:

$$\min_{\theta \in \mathbb{R}^d} \mathbb{E}_{(x,y) \sim p_{data}}[\max_{\|\tilde{x} - x\|_{\infty} \leq \epsilon} \ell(f_{\theta}(\tilde{x}, y))] \tag{23}$$

Let us call $T(x) := \tilde{x}$, thus, we have

$$\mathbb{E}_{(x,y) \sim p_{data}}[\max_{\|T(x) - x\|_{\infty} \leq \epsilon} \ell(f_{\theta}(T(x)y))] = \max_{T, \, \|T(x) - x\|_{\infty} \leq \epsilon, \, \forall x \in \mathcal{X}} \mathbb{E}_{(x,y) \sim p_{data}}[\ell(f_{\theta}(T(x), y))]]$$

By definition of the pushforward measure we have that

$$\mathbb{E}_{(x,y) \sim p_{data}}[\max_{\|T(x) - x\|_{\infty} \leq \epsilon} \ell(f_{\theta}(T(x)y))] = \max_{T, \, \|T(x) - x\|_{\infty} \leq \epsilon, \, \forall x \in \mathcal{X}} \mathbb{E}_{(\tilde{x},y) \sim T_{\sharp} p_{data}}[\ell(f_{\theta}(\tilde{x}, y))]]$$

which conclude our proof since then it is equivalent to maximize over $T$ or $p_{adv} = T_{\sharp} p_{data}$.  □

**Proposition 3.1.** *The DRO problem* (2) *can be relaxed by considering coupling measure. In that case the constraint set $\mathcal{P}$ in Equation 2 take the form:*

$$\mathcal{P}_{conv} = \{p_{adv} \; : \; \exists \pi \in \Pi(p_{data}|p_{adv}) \;\; with \;\; \mathrm{supp}(\pi) \subset \{(x, y, \tilde{x}, y) \; | \; \|\tilde{x} - x\|_{\infty} \leq \epsilon, \, y \in \mathcal{Y} \}\} \, .$$

*Proof of Proposition 3.1.* We will prove that

$$\mathcal{P} \subset \mathcal{P}_{conv} \tag{24}$$

and that $\mathcal{P}_{conv}$ is convex. First let us start by showing that,

$$\mathcal{P} \subset \mathcal{P}_{conv}. \tag{25}$$

Let us consider $p_{adv} \in \mathcal{P}$. By definition of $\mathcal{P}$ there exists $T : \mathcal{X} \to \mathcal{X}$ such that $p_{adv} = T_{\sharp} p_{data}$ and $\|T(x) - x\|_{\infty} \leq \epsilon$. Now, we can define the coupling measure $\pi_T := (I_d, I_d, T, I_d)_{\sharp} p_{data}$ where $(I_d, I_d, T, I_d)(x, y) = (x, y, T(x), y)$. We can verify that $\pi_T \in \Pi(p_{data}|T_{\sharp} p_{data})$ and that by definition of $T$, we have $\mathrm{supp} \, \pi_T \subset \{(x, y, \tilde{x}, y) \; \text{s.t.} \; \|\tilde{x} - x\|_{\infty} \leq \epsilon, \, x, x' \in \mathcal{X}, \, y \in \mathcal{Y} \}$.

Finally, let us prove that $\mathcal{P}_{conv}$ is convex. Let us consider $p_1, p_2 \in \mathcal{P}_{conv}$ and $\lambda \in [0, 1]$. By definition of $\mathcal{P}_{conv}$, we know that there exists $\pi_1 \in \Pi(p_{data}|p_1)$ and $\pi_2 \in \Pi(p_{data}|p_2)$ such that $\mathrm{supp}(\pi_1)$ and $\mathrm{supp}(\pi_2)$ are included in $\{(x, y, \tilde{x}, y) \; \text{s.t.} \; \|\tilde{x} - x\|_{\infty} \leq \epsilon, \, x, x' \in \mathcal{X}, \, y \in \mathcal{Y} \}$. By linearity of the marginalization we have that $\lambda \pi_1 + (1 - \lambda) \pi_2 \in \Pi(p_{data}|\lambda p_1 + (1 - \lambda) p_2)$. Moreoevr, we have that $\mathrm{supp}(\lambda \pi_1 + (1 - \lambda) \pi_2) = \mathrm{supp}(\pi_1) \cup \mathrm{supp}(\pi_2) \subset \{(x, y, \tilde{x}, y) \; \text{s.t.} \; \|\tilde{x} - x\|_{\infty} \leq \epsilon, \, x, x' \in \mathcal{X}, \, y \in \mathcal{Y} \}$. Thus we have $\lambda p_1 + (1 - \lambda) p_2 \in \mathcal{P}_{conv}$. It concludes the fact that $\mathcal{P}_{conv}$ is convex.  □

**Proposition 3.2** (convex relaxation of adversarial training)**.** *The convex DRO relaxation of adversarial training can be reformulated as*

$$\min_{\theta \in \mathbb{R}^d} \max_{p_{adv} \in \mathcal{B}_{\epsilon}(p_{data})} \mathbb{E}_{(\tilde{x}, y) \sim p_{adv}}[\ell(f_{\theta}(\tilde{x}, y)]$$

*where $\mathcal{B}_{\epsilon}(p_{data}) = \{p \mid W_{\infty, \infty}(p, p_{data}) \leq \epsilon\}$ and $W_{p, \infty}$ is defined in* (9).

*Proof of Proposition 3.2.* Under mild assumptions (see the assumption of [Villani, 2009, Prop 5.21]) $\mathcal{B}_{\epsilon}(p_{data}) = \{p \mid W_{\infty, \infty}(p, p_{data}) \leq \epsilon\} = \mathcal{P}_{conv}$.

$$p_{adv} \in \mathcal{P}_{conv} \iff \exists \pi \in \Pi(p_{data}|p_{adv}) \;\; \text{with} \;\; \mathrm{supp}(\pi) \subset \{(x, y, \tilde{x}, y) \; | \; \|\tilde{x} - x\|_{\infty} \leq \epsilon, \, y \in \mathcal{Y} \}$$

$$\tag{26}$$

$$\iff \exists \pi \in \Pi(p_{data}|p_{adv}) \;\; \text{with} \;\; \sup_{(z, \tilde{z}) \in \mathrm{supp}(\pi)} \|z - \tilde{z}\|_{\infty} \leq \epsilon \tag{27}$$

$$\iff \min_{\pi \in \Pi(p_{data}|p_{adv})} \sup_{(z, \tilde{z}) \in \mathrm{supp}(\pi)} \|z - \tilde{z}\|_{\infty} \leq \epsilon \tag{28}$$

Note that the second equivalence is possible because $\epsilon < \min_{y \neq y'} |y - y'|$ and the third one because the minimum is achieved by compactness of $\Pi(p_{data}|p_{adv})$ (see [Villani, 2009, Prop 5.21]).  □

**Proposition 3.3.** *If $\lambda \to \infty$, then $\pi_\lambda^* \to \pi^*$.*

*Proof of Proposition 3.3.* The proof is similar to the proof of Proposition 4.1 from Peyré and Cuturi [2019] since $\mathcal{P}$ is compact. □

**Proposition A.1.** *For any given $\theta \in \mathbb{R}^d$, the function $h_\theta(\pi) = \mathbb{E}_{(\tilde{x},y) \sim p_{adv}}[\ell(f_\theta(\tilde{x}), y)] + \frac{1}{\lambda} H(\pi)$ is strictly-concave and the constraints on $\mathcal{P}$ are convex.*

*Proof.* $\mathcal{P}$ and $\mathcal{F}$ are convex sets: both constraints of $\mathcal{P}$ are linear, and is convex by $\mathcal{F}$ by assumptions.

Moreover,

- $h_\theta(\pi)$ is the sum of a concave term and a strongly concave term. Indeed:
  - $\pi \to \mathbb{E}_{(\tilde{x},y) \sim p_{adv}}[\ell(f_\theta(\tilde{x}), y)]$ concave in $\pi$ by linearity.
  - $\pi \to \frac{1}{\lambda} H(\pi)$ is strongly concave in $\pi$.
- $\theta \to h_\theta(\pi)$ is convex by convexity of $l$ (e.g. mean squared loss, cross entropy loss, ...).

□

**Proposition 3.4.** *For any given $\theta \in \mathbb{R}^d$, The optimization problem* (12) *has a unique solution,*

$$\pi^*((x,y), (\tilde{x}, y)) \propto p_{data}(x,y) \exp(\lambda \ell(f_\theta(\tilde{x}), y)) \mathbf{1}_{\|x-\tilde{x}\|_\infty \leq \epsilon} \mathbf{1}_{\tilde{y}=y} . \tag{29}$$

*For a given natural example $x$ with a label $y$, $\pi(z, \tilde{z})$ grows with the propensity of $\tilde{x}$ to be better than the other elements of $\mathcal{B}_x(\epsilon)$ (the $\ell_\infty$-ball of radius $\epsilon$) to fool the classifier. We then deduce the explicit form of the adversarial distribution:*

$$p_{adv}^*(\tilde{x}, y) = \mathbb{E}_{x \sim p_{x,data}} [p_{data}(y|x) \frac{\exp(\lambda \ell(f_\theta(\tilde{x}),y)) \mathbf{1}_{\|x-\tilde{x}\|_\infty \leq \epsilon}}{\int_{\tilde{x} \in \mathcal{B}_\epsilon(x)} \exp(\lambda \ell(f_\theta(\tilde{x}),y)) d\tilde{x}}]$$

$$\pi^*(z, \tilde{z}) \propto p_{data}(x,y) \exp(\lambda \ell(f_\theta(\tilde{x}), y)) \mathbf{1}_{\|x-\tilde{x}\|_\infty \leq \epsilon} \mathbf{1}_{\tilde{y}=y} .$$

*Proof of Proposition 3.4.* The objective function is concave and at the optimum $\pi^*((x,y), (\tilde{x}, y)) > 0$ for each $(x, \tilde{x}, y)$ s.t. $\|x - \tilde{x}\| \leq \epsilon$. The objective function is $\mathcal{C}^1$ in the neighborhood of $\pi^*$. Thus, we can introduce dual variable $\alpha : \mathbb{R} \to \mathbb{R}$ s.t. $\nabla_\pi L(\pi^*, \alpha) = 0$. The Lagrangian function $L$ is:

$$L(\pi, \alpha) = \int_{\mathcal{Z} \times \mathcal{Z}} \ell(f_\theta(\tilde{x}), y) d\pi((x,y)(\tilde{x}, y)) - \frac{1}{\lambda} \int_{\mathcal{Z} \times \mathcal{Z}} (\log(\pi((x,y), (\tilde{x}, y)) - 1) d\pi((x,y)(\tilde{x}, y))$$
$$+ \int_{\mathcal{Z} \times \mathcal{Z}} \alpha(x,y) \left( p_{data}(x,y) dx - \int_{\mathcal{Z} \times \mathcal{Z}} d\pi((x,y)(\tilde{x}, y)) \right) \tag{30}$$

$$L(\pi, \alpha) = \int_{\mathcal{Z} \times \mathcal{Z}} (\ell(f_\theta(\tilde{x}), y) - \frac{1}{\lambda}(\log(\pi((x,y)(\tilde{x}, y))) - 1) - \alpha(x,y)) d\pi((x,y)(\tilde{x}, y))$$
$$+ \int_{\mathcal{Z}} \alpha(x,y) p_{data}(x,y) dx \tag{31}$$

$$\frac{\partial L(\pi, \alpha))}{\partial \pi} = \ell(f_\theta(\tilde{x}), y) - \frac{1}{\lambda} \log(\pi((x,y)(\tilde{x}, y))) - \alpha(x,y) = 0 \tag{32}$$

So at the optimum, for each $((x,y), (\tilde{x}, y))$ s.t. $\|x - \tilde{x}\| \leq \epsilon$, we have $\pi^*((x,y), (\tilde{x}, y)) = \exp(\lambda(\ell(f_\theta(\tilde{x}), y))) \exp(-\lambda \alpha(x,y)))$. In addition, we know

that $\int_{(\tilde{x},y)\in\mathcal{B}_\epsilon((x,y))} d\pi((x,y),(\tilde{x},y)) = p_{data}(x,y)$, thus $\exp(-\lambda\alpha^*(x,y)) = p_{data}(x,y)\frac{1}{\int_{\tilde{x}\in\mathcal{B}_\epsilon(x)}\exp(\lambda(\ell(f_\theta(\tilde{x}),y)))d\tilde{x}}$. Therefore, for each $((x,y),(\tilde{x},y))$ s.t. $\|x-\tilde{x}\| \le \epsilon$:

$$\pi^*((x,y),(\tilde{x},y)) = p_{data}(x,y)\frac{\exp(\lambda(\ell(f_\theta(\tilde{x}),y)))}{\int_{\tilde{x}\in\mathcal{B}_\epsilon(x)}\exp(\lambda(\ell(f_\theta(\tilde{x}),y)))d\tilde{x}}\mathbf{1}_{y=\tilde{y}}$$

And $\pi((x,y)(\tilde{x},y)) = 0$ for each $((x,y),(\tilde{x},y))$ s.t. $\|x-\tilde{x}\| > \epsilon$, i.e.:

$$\pi^*((x,y),(\tilde{x},y)) = p_{data}(x,y)\frac{\exp(\lambda(\ell(f_\theta(\tilde{x}),y)))}{\int_{\tilde{x}\in\mathcal{B}_\epsilon(x)}\exp(\lambda(\ell(f_\theta(\tilde{x}),y))d\tilde{x}}\mathbf{1}_{\|x-\tilde{x}\|\le\epsilon}\mathbf{1}_{y=\tilde{y}}$$

$$\pi^*((x,y)(\tilde{x},y)) \propto p_{data}(x,y)\exp(\lambda\ell(f_\theta(\tilde{x}),y))\mathbf{1}_{\|x-\tilde{x}\|_\infty\le\epsilon}\mathbf{1}_{y=\tilde{y}}.$$

Therefore, we have that:

$$\sup_\pi L(\pi,\alpha) = \frac{1}{\lambda} + \int_\mathcal{Z} \alpha(x,y)p_{data}(x,y)d(x,y) \tag{33}$$

$$\sup_\pi L(\pi,\alpha) = \frac{1}{\lambda} - \frac{1}{\lambda}\int_\mathcal{Z} p_{data}(x,y)\log(\frac{p_{data}(x,y)}{\int_{\tilde{x}\in\mathcal{B}_\epsilon(x)}\exp(\lambda(\ell(f_\theta(\tilde{x}),y)))d\tilde{x}})d(x,y) \tag{34}$$

$$\sup_\pi L(\pi,\alpha) = \frac{1}{\lambda}\left(\int_\mathcal{Z} p_{data}(z)\log(\int_{\tilde{x}\in\mathcal{B}_\epsilon(x)}\exp(\lambda(\ell(f_\theta(\tilde{x}),y)))d\tilde{x}))dz - \int_\mathcal{Z} p_{data}(z)\log(p_{data}(z))dz + 1\right) \tag{35}$$

$\square$

**Proposition 3.5.** *Let us call* $\mathbb{E}_{(\tilde{x},y)\sim p_{adv}}[\ell(f_\theta(\tilde{x}),y)] + \frac{1}{\lambda}H(\pi) = \varphi(\theta,\pi)$. *If* $\{f_\theta \mid \theta \in \Theta\}$ *is a convex set, then we have that*

$$\min_{\theta\in\mathbb{R}^d}\max_{\pi\in\mathcal{P}}\varphi(\theta,\pi) = \max_{\pi\in\mathcal{P}}\min_{\theta\in\mathbb{R}^d}\varphi(\theta,\pi) \tag{36}$$

*where* $\mathcal{P} = \{p_{adv} : \exists\pi \in \Pi(p_{x,data}|_x, p_{x,adv})$ *with* $\pi((z),(\tilde{x},y)) = 0, \forall x,\tilde{x} \in \mathcal{X}$ *s.t.* $\|\tilde{x} - x\|_\infty > \epsilon\}$.

*Proof.* The proof is similar to the proof of Theorem 28 from Merigot and Thibert [2020]. $\square$

**Proposition 3.6.** *Let us assume that the datapoints* $x_i$ *are independently sampled from a distribution* $p$ *over* $[0,1]^d$ *that is absolutely continuous with respect to Lesbegue measure and has a upper-bounded value, i.e.* $p(x) \le M$, $x \in [0,1]^d$. *Then* $\mathbb{P}(\exists i \ne j \in [n], \|\tilde{x}_j - x_i\|_\infty \le \epsilon) = O(n^2\epsilon^d)$.

*Proof of Proposition 3.6.* Let us consider the quantity

$$\mathbb{P}(\exists i \ne j \in [n], \|\tilde{x}_j - x_i\|_\infty \le \epsilon) \le \frac{n(n-1)}{2}\mathbb{P}(\|\tilde{x} - x\|_\infty \le \epsilon) \tag{37}$$

$$\le \frac{n(n-1)}{2}MVol(\mathcal{B}(x,\epsilon)) \tag{38}$$

$$= \frac{n(n-1)}{2}M\epsilon^d \tag{39}$$

$\square$

**Theorem 3.2** (Characterization of the solutions in the semi discrete case). *For any $\lambda > 0$, the solution set of the regularized adversarial training problem* (11) *is:*

$$\theta^* \in \arg\min \frac{1}{n\lambda} \sum_{i=1}^{n} \log\left( \int_{||\tilde{x}-x_i|| \leq \epsilon} \exp(\lambda \ell(f_\theta(\tilde{x}), y_i)) d\tilde{x} \right) =: \arg\min g(\theta), \qquad (40)$$

*and under Assumption 3.1 the optimal adversarial distribution associated with this adversarial training problem is*

$$p_{adv}(\tilde{x}, y) = \begin{cases} \dfrac{\exp(\lambda \ell(f_\theta(\tilde{x}), y))}{n \int_{\tilde{x} \in \mathcal{B}_\epsilon(x_i)} \exp(\lambda \ell(f_\theta(\tilde{x}), y)) d\tilde{x}} & \text{if } \exists i \in [n] \text{ s.t. } \|x_i - \tilde{x}\|_\infty \leq \epsilon \\ 0 & \text{otherwise.} \end{cases} \qquad (41)$$

*Proof of Theorem 3.2.*

$$\min_{f_\theta \in \mathcal{F}} \max_{p_{adv} \in \mathcal{P}} \mathbb{E}_{(x, \tilde{x}, y) \sim \pi(z, (\tilde{x}, y))} [\ell(f_\theta(\tilde{x}), y)] + \frac{1}{\lambda} H(\pi) \qquad (42)$$

From Proposition 3.5:

$$= \min_{f_\theta \in \mathcal{F}} \left( \min_\alpha \left( \int_\mathcal{X} \alpha(x, y) p_{data}(x, y) dx \right) + \frac{1}{\lambda} \right) \text{ for } \alpha(x, y) = -\frac{1}{\lambda} \log\left( \frac{p_{data}(x, y)}{\int_{\tilde{x} \in \mathcal{B}_\epsilon(x)} \exp(\lambda(\ell(f_\theta(\tilde{x}), y)))) d\tilde{x}} \right) \qquad (43)$$

$$= \min_{f_\theta \in \mathcal{F}} \frac{1}{\lambda} \left( \int_\mathcal{X} p_{data}(x, y) \log\left( \int_{\|x-\tilde{x}\| \leq \epsilon} \exp(\lambda(\ell(f_\theta(\tilde{x}), y))) d\tilde{x} \right) dx - \int_\mathcal{X} p_{data}(x, y) \log(p_{data}(x, y)) dx + 1 \right) \qquad (44)$$

$$\square$$

Finally, the probability distribution of the adversarial examples $p_{adv}$ is written:

$$p_{adv}(\tilde{x}, y) = \int_{x \in \mathcal{X}} d\pi((x, y, )(\tilde{x}, )) \qquad (45)$$

$$p_{adv}(\tilde{x}, y) = \int_\mathcal{X} p_{data}(x, y) \frac{\exp(\lambda \ell(f_\theta(\tilde{x}), y))}{\int_\mathcal{X} \exp(\lambda \ell(f_\theta(\tilde{x}), y)) d\tilde{x}} 1_{\|x-\tilde{x}\| \leq \epsilon} dx \qquad (46)$$

$$p_{adv}(\tilde{x}, y) = \mathbb{E}_{x \sim p_{x, data}} [p_{data}(y|x) \frac{\exp(\lambda \ell(f_\theta(\tilde{x}), y)) 1_{\|x-\tilde{x}\|_\infty \leq \epsilon}}{\int_{\tilde{x} \in \mathcal{B}_\epsilon(x)} \exp(\lambda \ell(f_\theta(\tilde{x}), y))}] \qquad (47)$$

In practice: $\forall i \quad p_{data_i} = \frac{1}{n}$, so we have:

$$p_{adv}(\tilde{x}, y) = \begin{cases} \dfrac{\exp(\lambda \ell(f_\theta(\tilde{x}), y))}{n \int_{\tilde{x} \in \mathcal{B}_\epsilon(x_i)} \exp(\lambda \ell(f_\theta(\tilde{x}), y))} & \text{if } \exists i \in [n] \text{ s.t. } \|x_i - \tilde{x}\|_\infty \leq \epsilon \\ 0 & \text{otherwise.} \end{cases}$$

and then,

$$\theta^* \in \arg\min \frac{1}{n\lambda} \sum_{i=1}^{n} \log\left( \int_{||\tilde{x}-x_i|| \leq \epsilon} \exp(\lambda \ell(f_\theta(\tilde{x}), y_i)) d\tilde{x} \right) =: \arg\min g(\theta),$$

# B ADDITIONAL RESULTS

## B.1 TUNING OF HYPERPARAMETERS

In this section, we detail the tuning of different hyperparameters. Among them:

- $T$ : The number of Langevin iterations for crafting adversaries during the training process.
- $\sigma$: Represents the noise which is the particularity of the Langevin. Indeed, it differs from the gradient descent by the addition of Gaussian noise. The noise allows adding randomness in the crafting of adversarial examples. Thus, the size of the noise, as well as its shape, are of primary importance in the performance of Langevin. For example, a too important noise will generate totally random examples and will erase the weight of the gradient.
- $\eta_l = \lambda \times \gamma$: represents the learning rate of Langevin MC. The parameter represents the learning rate of Langevin.

In general, even if $T \geq 2$, we only keep the last batch of adversarial examples for updating $\theta$. Here, all the results are presented for $T = 1$ which gives the best results and the best computation time. A further study could analyze the influence of using more batches of adversarial examples at each epoch of the training. The results for the influence of $\sigma$ and $\eta_l(\lambda)$ are provided below:

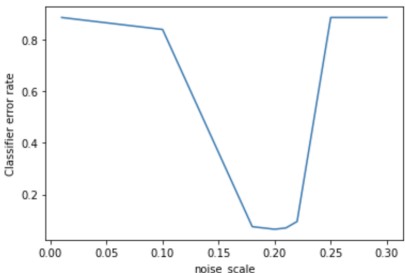

Figure 3: Influence of $\sigma$ (noise_scale) on the performance of the classifier when training with Adversarial Transport framework with architecture A. These results are for $T = 1$ and $\eta_l(\lambda) = 0.2$. We did the same experiment with other values of $\eta_l(\lambda)$ but whatever the value of $\sigma$, $\eta_l(\lambda) = 0.2$ gives the best result.

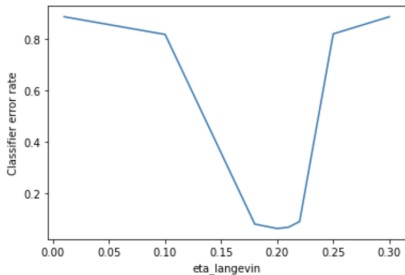

Figure 4: Influence of $\eta_l(\lambda)$ (eta_langevin) on the performance of the classifier when training with Adversarial Transport framework with architecture A. These results are for $T = 1$ and $\sigma = 0.2$. We did the same experiment with other values of $\sigma$ but whatever the value of $\eta_l(\lambda)$, $\sigma = 0.2$ gives the best result.

The two curves are similar. We notice that the method converges only for small range of values of these hyperparameters. Regarding that $\lambda = \frac{2\eta_l}{\sigma^2}$, these experiments show that the best value for $\lambda$ we can take is $\lambda = 10$. These graphs justify the fact that our whole study was conducted with $\eta_l(\lambda) = 0.2$ and $\sigma = 0.2$.

### B.2 MODELS ARCHITECTURE

We study the influence of the architecture of the neural network on the performance of our model. We test our algorithm with the two models below. Model B is the one used by Madry et al. [2018] for the PGD framework.

| Architecture A |
| --- |
| Conv(1,64,5) |
| ReLU() |
| Conv(64,64,5) |
| ReLU() |
| Dropout(0.25) |
| Linear($64 \times 20 \times 20$, 128) |
| Dropout(0.5) |
| Linear(128,10) |

| Performance | Langevin | PGD |
| --- | --- | --- |
| Train Adversarial | 98.48% | 94.81% |
| Clean test set | 99.16% | 98.82% |
| FGSM | 96.53% | 94.95% |
| PGD 40 | 93.12% | 92.28% |
| PGD 100 | 91.87% | 91.18% |

Table 3: Comparison of the performance of robust classifiers trained with the Adversarial Transport framework (Langevin) and PGD (trained with 40 iterations). The architecture of the two classifiers is the above architecture A. The robustness of these two models is tested against FGSM attacks, PGD with 40 iterations (PGD 40) and PGD with 100 iterations (PGD 100). The trained classifier with Langevin is more robust than the one with PGD.

| Architecture B |
| --- |
| Conv(1,32,3,padding = 1) |
| ReLU() |
| Conv(32,64,3, padding = 1, stride = 2) |
| ReLU() |
| Conv(32,64,3, padding = 1) |
| ReLU() |
| Conv(32,64,3, padding = 1) |
| ReLU() |
| Flatten() |
| Linear($7 \times 7 \times 64$, 100) |
| ReLU() |
| Linear(100,10) |

| Performance | Langevin |
| --- | --- |
| Train Adversarial | 99.42% |
| Clean test set | 99.10% |
| FGSM | 97.73% |
| PGD 40 | 94.82% |
| PGD 100 | 94.18% |

Table 4: Performance of robust classifiers trained with the Adversarial Transport framework (Langevin) with architecture B. The robustness of this model is tested against FGSM attacks, PGD with 40 iterations (PGD 40) and PGD with 100 iterations (PGD 100). The results are better than for architecture A. Especially, we notice that there is no loss of performance between PGD 40 attacks and PGD 100 attacks. As explained in Madry et al. [2018], networks with larger capacity are stronger against adversarial attacks.

| Performance | Langevin | PGD 40 |
| --- | --- | --- |
| Train Adversarial | 98.90% | 97.42% |
| Clean test set | 84.07% | 83.54% |
| FGSM | 56.47% | 59.51% |
| PGD 40 | 55.64% | 58.88% |
| PGD 100 | 54.50% | 57.97% |
| Autoattack | 26.12% | 26.04% |

Table 5: Performance comparison of a wide ResNet (Zagoruyko and Komodakis [2017]) trained with the Adversarial Transport framework (Langevin) and PGD 40 on CIFAR-10. No hyperparameter tuning was done; the same setup as described in Section 5 is used with the exception of only training for 36 epochs instead of 100. The Langevin framework took about 2h10m to train, while PGD 40 took 43h43m to train.

