# OpenReview forum: "A Distributional Robustness Perspective on Adversarial Training with the $\infty$-Wasserstein Distance"
_ICLR.cc/2022/Conference — ICLR 2022 Submitted_

### Official Review · Reviewer_mcjf · 2021-11-02

**Correctness:** 4
**Technical Novelty And Significance:** 1
**Empirical Novelty And Significance:** 1
**Recommendation:** 5
**Confidence:** 4

**Main Review:**

The strength of this paper is a new adversarial training method that treats the problem as distributional-wise, as compared with traditional sample-wise adversarial training methods, and the proposed method has been shown to have better performance than some baselines.

The main weakness is the novelty of this paper. The technical or algorithmic contribution of this paper seems to be very limited.
The equivalence between the adversarial training under infinity-norm perturbations and the DRO under \infty-\infty-Wasserstein distance is very straightforward. Moreover, given such equivalence, the paper does not provide enough motivation about why we still want to consider the distributionally adversarial training. Also, instead of solving the DRO problem, the entropic regularizer is imposed in order to get a closed-form solution of the least favorable distribution. And it would be good to provide more theoretical insights.
Also, in the numerical results, how about the training time comparison between the proposed method and baseline methods?

Typoes and minor issues:
1 Page 6: “The latter point can be supported that the following…” should be “supported by the following”
2 Page 7: the Lesbegue measure $q(\tilde x)$ should be proportional to $exp(-U(\tilde x))$
3 Algorithm 1 line 9, should be $\tilde{x}^{t}$ on the right hand side
4 Figure 1 and 2: It would be better to use same color to denote the same method.

**Summary Of The Paper:**

This paper establishes the equivalence between the adversarial training problem (under infinity-norm) and the Distributionally robust optimization (DRO) problem with \infty-\infty-Wasserstein distance. Then they propose to use an entropic regularizer to find the regularized optimal adversarial distribution more efficiently. Numerical results on MNIST and CIFAR-10 validate the performance of the proposed method.

**Summary Of The Review:**

In summary, although this paper show certain equivalences between adversarial training and DRO, it is mostly a straightforward result under the infinity norm, and the overall contribution of this paper seems to be limited.

---

> ### Author Response · Authors · 2021-11-16
> **Response to mcjf**
>
> We would like to thank reviewer mcjf for the insightful comments. Below we address the concerns and questions mentioned by reviewer mcjf:
> - Concerning the algorithmic contribution of the paper :
>   - “The technical or algorithmic contribution of this paper seems to be very limited.” We respectfully disagree.
> The algorithmic contributions of our paper are significant. Our paper provides a tractable algorithm based on new  theoretical results.
>   - In addition, our algorithm is better than the SOTA (Projected Gradient Descent adversarial training) methods. Indeed, confronted with the most robust attacks (PGD attacks with 100 iterations) our framework is about 6% more robust than the SOTA model.
>   - On top of that, our algorithm provides a speed-up compared to the state-of-the art of respectively x200 for MNIST and x8 for CIFAR-10 (see figure 1).
>   - Our algorithm is even faster than PGD with 1 iteration thanks to the fact that it restarts at each time step from the adversarial example generated at the previous step, so no need to recalculate the adversarial example at each step (cf paragraph reusing the examples).
> - Concerning the technical contribution of the paper :
> We respectfully disagree as well.
>   - First of all,  the equivalence between the adversarial training under infinity-norm perturbations and the DRO under $\infty-\infty$-Wasserstein distance was never used before.
>   - From there, our paper our work provides a closed form formulation of the least favorable adversarial distribution, which is new and then proposes a tractable approach.
>   - As explained in Introduction and Related Work, working at the scale of distribution enables our classifier to become robust against a wide range of attacks and which is confirmed, as explained above, by our numerical results.
> - Typoes and minor issues: Thank you for your comments, we did an update with your recommandations.
> - "How about the training time comparison between the proposed method and baseline methods?"
> It is complicated to compare training time because it is hardware dependent, software dependent and implementation dependent, that is why it is safer and more relevant to compare the number of backpropagations (which is a proxy for time).
> - Theoretical insights about the entropic regularizer:
> Theoretical insights are given section 4.2 reusing examples (page 8): ”This idea is justified by the fact that the distribution $p_{adv}(\tilde{x},y) \propto \exp(\lambda \ell (f_\theta(\tilde x),y))$ is actually a continuous function of $\theta$. It can be seen with duality argument from eq(12) where the function $h_\theta(\pi)$ is strictly concave with respect to $\pi$ which implies the continuity of $\theta \mapsto \arg \max_\pi h_\theta(\pi)$. We leave as a future work the formalization of this argument.”

---

> > ### Comment · Reviewer_mcjf · 2021-11-29
> > **Thanks**
> >
> > I thank the authors for the extensive answer and appreciate the time they took! However, after reading the response and other reviewers' comments, my rating remains unchanged.

---

### Official Review · Reviewer_CrMc · 2021-11-03

**Correctness:** 3
**Technical Novelty And Significance:** 3
**Empirical Novelty And Significance:** 1
**Recommendation:** 5
**Confidence:** 4

**Main Review:**

**Strengths of the paper:**
- The technical contribution of this paper is significant because it shows a connection between standard adversarial training such as PGD and distributional robustness.
-  All obtained theoretical results are meaningful and have clear impacts.

**Weaknesses of the paper:**
- Although the obtained theoretical results sound reasonable and meaningful, the mathematical communication and proof in this paper seem not solid to me.  Specifically, in all derivations in Appendix, it only mentions $\pi(x, x')$ with a lack of the label $y$. Still, it is unclear how to associate label y to Eq. (11) since Eqs. (9,10) only mention $x,x'$ and a WS distance between data distributions.
- There are no proofs of Theorems 3.1 and 3.2 in Appendix. Moreover, Appendix is not well-organized to connect to the main paper.
-  The experiments are too humble which need a comparison with more baselines using more advanced architectures. Moreover, the authors should report the robust accuracy for black-box attacks and Auto-Attack.

**Summary Of The Paper:**

- This paper proposes using $\infty$-Wasserstein distributional robustness for improving model robustness. Specifically, to obtain a tractable solution, it adds an entropic regularization term to the primal form of $\infty$-Wasserstein distributional robustness. The Langevin algorithm is used to sample adversarial examples for adversarial training.

- Although the experimental results show the merit of the proposed method, the experiments are still humble with a lack of experiments on more impactful architectures such as WideResNet and comparison to other SOTA adversarial training methods.

**Summary Of The Review:**

This is a technically significant paper because it shows the connection between $\infty$-Wasserstein distributional robustness and standard adversarial training PGD. All obtained theoretical results are meaningful and have clear impacts. However, the presentation and writing of the paper need to be improved to make it a more solid paper.
- How to associate the label $y$ to the formulation?
-  Appendix needs to be organized better and all proofs need to be consistent with the main paper.

The experiments are still humble which need a comparison with more baselines using more advanced architectures. Moreover, the robust accuracy for black-box attacks and Auto-Attack should be reported.

It is unclear why the proposed method is faster than PGD even a single adversarial particle is searched using the Langevin algorithm.

---

> ### Author Response · Authors · 2021-11-16
> **Response to CrMc**
>
> We would like to thank reviewer CrMc for the insightful comments. Below we address the concerns and questions mentioned by reviewer CrMc:
> - Weaknesses of the paper:
>   -  We updated $\pi(x,\tilde{x})$ into  $\pi(z,\tilde{z})$ (with $z = (x,y)$ and $\tilde{z}=(\tilde{x},y)$).
>   - "Still, it is unclear how to associate label $y$ to Eq. (11) since Eqs. (9,10) only mention $x$,$\tilde{x}$" : $y$ is the true label, then the same $y$ is attributed to $x$ and $\tilde{x}$. So, the distance between $x$ and $\tilde{x}$ is not related to the label $y$, while in eq(11), $y$ has a major influence because it quantifies the error between the classification of the adversarial example $\tilde{x}$ and the true label $y$. We did some modification and replaced $\pi(x,\tilde{x})$ by  $\pi((x,y),(\tilde{x},y))$ and we hope it is clearer now.
>   - We re-organized the appendix. We add proofs for propostions 3.1 and 3.2. Is it clearer now?
>   - Do you have any recommendations for baselines and advanced architectures? We should have robust accuracy for black-box attacks and Auto-Attack before the end of the rebuttal.
> - "It is unclear why the proposed method is faster than PGD even a single adversarial particle is searched using the Langevin algorithm" :
> The advantage of our algorithm is that it restarts at each time step from the adversarial example generated at the previous step, so no need to recalculate the adversarial example at each step (cf paragraph reusing the examples).

---

> > ### Comment · Reviewer_CrMc · 2021-11-25
> > **Additional questions**
> >
> > Thanks for your clarification. I have some further questions.
> >
> > 1. What is $\Pi(p_{data} \mid p_{adv})$ in the definition of $P_{conv}$? It seems that this notion is not used in theoretical development.
> >
> > 2. There is a gap between the DRO formula in Eq. (10) and the optimization in Eq. (11). I believe that you start from Eq. (10), but it needs a derivation from Eq. (10) to Eq. (11) which is based on to develop the practical approach.
> >
> > 3. The derivation from Eq. (9) to Eq. (10) is not clear to me. The reason is that $z= (x,y) \sim p_{data}$ and $z' = (x', y') \sim p_{adv}$, but the authors never mention to how to define the metric between $z$ and $z'$. It is worth mentioning because $y$ and $y'$ are two discrete labels.

---

> > > ### Author Response · Authors · 2021-11-29
> > > **Answer to Additional questions**
> > >
> > > Hello,
> > > 1)  $\Pi(p_{data},p_{adv})$ is the space of transport plan between $p_{data}$ and $p_{adv}$ (cf Definition 3.1)
> > > 2) The only difference between Eq.(10) and Eq.(11) is that we add the entropic regularization term in Eq.(11).
> > > 3) We can define a metric the following ways:
> > >  - $\|z-z'\| = \|x-x'\| + |y-y'| $ (cf footnote page 4)
> > >  - Otherwise, we can also consider $\|z-z'\| = \|x-x'\| + 1_{y=y'}$
> > >
> > > The fact is that $\|z-z'\| > \epsilon$ if $y \neq y' $.
> > >  Is it clearer now?

---

> > > > ### Comment · Reviewer_CrMc · 2021-11-29
> > > > **Feedback to your answers**
> > > >
> > > > Thanks for your answer.
> > > > 1) $\Pi(p_{data}, p_{adv})$ is the space of transport plans or couplings, but you denoted as $\Pi(p_{data} \mid p_{adv})$, so it is not consistent. In addition, $P_{conv}$ was not used anywhere.
> > > > 2) I cannot see $|z - z'| = |x-x'| + |y-y'|$ in the footnote at Page 4. Although the assumption is necessary, you need to clearly define $|z-z'|$.
> > > > 3) Eq. (10) searches for $p_{adv}$, while Eq. (11) searches for the coupling $\Pi$. Therefore, there is a gap that needs more explanations there.

---

> > > > > ### Author Response · Authors · 2021-11-30
> > > > > **Answer to Feedback to previous answers**
> > > > >
> > > > > Thank you for your comments. We will take them into account.

---

> ### Author Response · Authors · 2021-11-23
> **Results for auto-attack**
>
> Hello,
>
> We add the results for auto-attacks for MNIST (see table 1 in the paper).
> We have :
> - Langevin (our algorithm) 88.55% accuracy
> - PGD (with 40 iterations) : 89.10% accuracy

---

### Official Review · Reviewer_qU23 · 2021-11-03

**Correctness:** 3
**Technical Novelty And Significance:** 2
**Empirical Novelty And Significance:** 2
**Recommendation:** 5
**Confidence:** 4

**Main Review:**



The results linking adversarial robustness to distributional robustness in Section 2 are both presented without proof, and already known and published. There are quite a few papers that present similar results, for example:
https://arxiv.org/pdf/1710.10016.pdf
http://proceedings.mlr.press/v139/cranko21a/cranko21a.pdf

A significant weakness of this submission is the mathematical rigour, which is sorely lacking. Many of the propositions are presented without proofs, and there are simple mistakes of measure theory present throughout. For example on page 3, $\pi$ is declared to be a coupling on the product $\mathcal X \times \mathcal X$, yet later $\pi:\mathcal X \times \mathcal X \to [0,+\infty]$ is written. When this should in fact be $\pi:\mathcal B(\mathcal X \times \mathcal X) \to [0,+\infty]$. Classically couplings are required to also be probability distributions, so this should in fact be  $\pi:\mathcal B(\mathcal X) \otimes \mathcal B (X) \to [0,1]$. This is an aesthetic issue, but in Proposition 3.1 it has real significance since the constraint $\pi(x,\tilde x) = 0$ is meaningless. A proof of this proposition is not presented and so it is impossible to evaluate the rigour here. The same issue occurs through the proof of Lemma A.2, where, more egregiously $\log ( \pi(x,x') )$ is integrated without a treatment of infinities.

I believe it would substantially improve the paper to provide a proper discussion of the literature and characterising how these results improve upon existing ones. As an editorial note, if a proposition or theorem is simple enough to present without proof then it should not be leveraged as a cornerstone result, or perhaps included at all.

The experimental results however are encouraging, in the time since submission have you been able to more thoroughly
evaluate your model against other state of the art approaches?




**Summary Of The Paper:**

The authors present a connection between adversarial robustness and distributional robustness. A duality result is given linking the $\infinity$-Wasserstein distance to an entropy regularisation problem. The problem is then solved using a Langevan-sampling based algorithm.




**Summary Of The Review:**

The authors present an encouraging algorithm that appears to perform well. However, section 2 seems to consist almost entirely of known results. The new theoretical results included however are often shown without proof and are lacking in rigour.

---

> ### Author Response · Authors · 2021-11-16
> **Response to qU23**
>
> We would like to thank reviewer qU23 for the insightful comments. Below we address the concerns and questions mentioned by reviewer qU23:
>
>  - Paper with similar results:
>
> Both of the papers mentioned are relevant and there is indeed a link with our work. However even if http://proceedings.mlr.press/v139/cranko21a/cranko21a.pdf do use regularizer and distributional robustness framework, our perspective is significantly different. Indeed, the previously mentioned paper doesn’t use the link between DRO and adversarial training. They do not derive a closed-form solution for the optimal adversarial distribution nor a tractable algorithm. Concerning https://arxiv.org/pdf/1710.10016.pdf, it does not study adversarial training.
> - Weaknesses of the paper:
>
>   - Mathematical rigor. Thank you for your remark about the coupling $\pi$. Indeed, in our paper, $\pi$ is a probability distribution  $\pi \in \mathcal{B}(\mathcal{X})\otimes\mathcal{B}(\mathcal{X}) \to [0,1]$.
>   - "This is an aesthetic issue, but in Proposition 3.1 it has real significance since the constraint  $\pi(x,\tilde{x})$ is meaningless".  $\pi((x,y),(\tilde{x},y))=0$ when $\|x-x'\| \leq \epsilon$, i.e. $\pi$ support is $supp(\pi) \subset ((x,y,\tilde{x},y),
>     \|\tilde{x} -x\|_\infty \leq \epsilon, y \in Y )$. We made the updates in the paper. Do you find it clearer now? "More egregiously $\log⁡(\pi(x,\tilde x))$ is integrated without a treatment of infinities." $\log⁡(\pi(x,\tilde x))$ is integrated over the support of $\pi$, the resulting calculation is therefore rigorous.
>   - We added demonstrations of Proposition 3.1 and 3.2 in the appendix.
>
> - "I believe it would substantially improve the paper to provide a proper discussion of the literature and characterise how these results improve upon existing ones".
>
> We believe, we have already discussed why our paper presents significant theoretical improvements in the paragraph “Related Work”. We provide papers and works that led us to make the parallel between $\infty-\infty$-Wasserstein and adversarial training and explain why our framework is new and provides significant contributions.
>
> - "The experimental results, however, are encouraging, in the time since submission have you been able to more thoroughly evaluate your model against other state of the art approaches?"
>
> Do you have any recommendations for other meaningful state-of-the-art approaches?
> We should have robust accuracy for black-box attacks and Auto-Attack before the end of the rebuttal.

---

> > ### Comment · Reviewer_qU23 · 2021-11-18
> > **Response**
> >
> > My comment was not merely that there exist papers with the same result, but rather that the results presented in Section 2 are not a significant contribution.
> >
> > Thank you for including proofs. The improvement to the statement of Proposition 3.1is clearer. However the proofs you have provided are incomplete. For example you write
> > "By definition of $\mathcal{P}$ there exists $T:\mathcal X \to \mathcal X$ such that ...", the existence of the mapping needs to be shown. The existence of a Monge transportation map is nontrivial and requires assumptions made on the distributions involved.
> >
> > Regarding the logarithm, in Equation 30, you write $\log(\pi(z, \tilde z))$. Ignoring my previous comment that $\pi$ is a distribution and the meaning of (30) is not even clear mathematically, if we integrate the logarithm of a function that can take arbitrary small positive values, the integral may not be finite. These are all technical matters that can be dealt with, but your paper ignores them.

---

> > > ### Author Response · Authors · 2021-11-18
> > > **Follow-up on your response**
> > >
> > > Hello,
> > >
> > > Thank you for following-up
> > >
> > > On the significance of the results in Section 2: We do not claim that they are the main contribution of our paper but these results are necessary to build our theory and our algorithms.
> > >
> > >
> > > Regarding $p_{adv}$:
> > > We are *not* in the standard optimal transport setting where $p_{adv}$ and $p_{data}$ would be fixed in advance (in this case, we agree that Monge transportation maps may not exist).
> > > The set $\mathcal P$ is the set of push forward measures of the form $T_\sharp p_{data}$ so by definition any $p_{adv} \in \mathcal{P}$ is the result of a pushforward of the data distribution.
> > > In short, we only consider the set of $p_{adv}$ for which a transportation map exists (with an additional constraint on the transportation map that ensures that the resulting distribution correspond to adversarial examples)
> > >
> > > On the logarithm:
> > > Regarding Equation (30) and all our theory in Section 3.3. We assume that $p_{data}$ admits a density with respect to the Lebesgue measure. Thus we look for a $\pi^*$ that also admits a density. In that situation (when $\pi$ admits a density) the entropy is well defined since $\pi \log \pi$ is defined in 0.
> > >
> > > Thank you for calling out this lack of rigor we will clarify this in the revision of our paper by the end of the revision period.

---

### Decision · Program_Chairs · 2022-01-20

**Decision:**

Reject

**Comment:**

In this paper, authors study adversarial examples from a distributional robustness point of view. Reviewers had several concerns about the work and all thought the paper is not above the accept threshold. In particular, they mentioned that the presentation and writing of the paper need to be improved and results (specially the ones presented in Section 2) are not significant contributions and novel. Given all, I think the paper needs more work before being accepted.